# Review of Immune-Related Adverse Events (irAEs) in Non-Small-Cell Lung Cancer (NSCLC)—Their Incidence, Management, Multiorgan irAEs, and Rechallenge

**DOI:** 10.3390/biomedicines10040790

**Published:** 2022-03-28

**Authors:** Raju Vaddepally, Rajiv Doddamani, Soujanya Sodavarapu, Narasa Raju Madam, Rujuta Katkar, Anupama P. Kutadi, Nibu Mathew, Rohan Garje, Abhinav B. Chandra

**Affiliations:** 1Yuma Regional Medical Center, 2400 S Avenue A, Yuma, AZ 85364, USA; nmadam@yumaregional.org (N.R.M.); rkatkar@yumaregional.org (R.K.); akutadi@yumaregional.org (A.P.K.); nmathew@yumaregional.org (N.M.); abchandra@yumaregional.org (A.B.C.); 2Slidell Memorial Hospital, 1001 Gause Blvd, Slidell, LA 70458, USA; rajiv.doddamani@apogeephysicians.com; 3San Joaquin General Hospital, 500 W Hospital Road, French Camp, CA 95231, USA; ssodavar@sjgh.org; 4Department of Internal Medicine-Hematology/Oncology, University of Iowa, Iowa, IA 52242, USA; rohan-garje@uiowa.edu

**Keywords:** NSCLC, immunotherapy, irAEs, rechallenge, multisystem irAEs

## Abstract

Immune checkpoint inhibitors (ICIs) have revolutionized the treatment of advanced malignancies, including non-small cell lung cancer (NSCLC). These agents have improved clinical outcomes and have become quite an attractive alternative alone or combined with other treatments. Although ICIs are tolerated better, they also lead to unique toxicities, termed immune-related adverse events (irAEs). A reconstituted immune system may lead to dysregulation in normal immune self-tolerance and cause inflammatory side effects (irAEs). Although any organ system can be affected, immune-related adverse events most commonly involve the gastrointestinal tract, endocrine glands, skin, and liver. They can occur anytime during the treatment course and rarely even after completion. Owen and colleagues showed that approximately 30% of patients with NSCLC treated with ICIs develop irAEs. Kichenadasse et al. conducted a thorough evaluation of multiorgan irAEs, which is of particular interest because information regarding these types of irAEs is currently sparse. It is important to delineate between infectious etiologies and symptom progression during the management of irAEs. Close consultation with disease-specific subspecialties is encouraged. Corticosteroids are the mainstay of treatment of most irAEs. Early intervention with corticosteroids is crucial in the general management of immune-mediated toxicity. Grade 1–2 irAEs can be closely monitored; hypothyroidism and other endocrine irAEs may be treated with hormone supplementation without the need for corticosteroid therapy. Moderate- to high-dose steroids and other additional immunosuppressants such as tocilizumab and cyclophosphamide might be required in severe, grade 3–4 cases. Recently, increasing research on irAEs after immunotherapy rechallenge has garnered much attention. Dolladille and colleagues assessed the safety in patients with cancer who resumed therapy with the same ICIs and found that rechallenge was associated with about 25–30% of the same irAEs experienced previously (4). However, such data should be carefully considered. Further pooled analyses may be required before we conclude about ICIs’ safety in rechallenge.

## 1. Introduction

Non-small-cell lung cancer (NSCLC) is the commonly reported variant (85%) of lung cancer with a 5-year survival rate of 15%. Although practical therapeutic approaches in NSCLC, including cytoreductive surgery, chemotherapy, and radiotherapy, substantially increase the survival time, the overall prognosis remains unchanged likely due to the tumor heterogeneity and increased mutations. Recently, immunotherapy with immune checkpoint inhibitors (ICIs) has revolutionized the therapeutic management of NSCLC [1,2].

Evading immunity is a hallmark of cancer cells. Various immunotherapeutic modalities aim to target the evasion potential of cancer cells by harnessing the power of the immune system. One such crucial strategy includes ICIs. The ICIs trigger numerous diminutive mechanisms such as inhibiting T-cell apoptosis, abating peripheral T effector cell exhaustion, and facilitating the conversion of T cells to regulatory T cells (Treg cells), enhancing T-cell antitumor activity and eventually leading to immune control and death of the cancerous cells [3,4]. However, any organ system can be affected due to the immune system’s nonspecific activation by the immune checkpoint inhibitors leading to irAEs.

The underlying pathophysiology of irAEs mimics that of an autoimmune process. It is crucial to understand their uniqueness compared to chemotherapy that stems from their mechanism of action. ICIs now have an increasing number of indications, including but not limited to neoadjuvant settings, combination therapy with chemotherapy, radiotherapy, or targeted agents. Increased incidence of irAEs can be attributed due to the steady rise in ICI therapy indications and popularity. It is essential for physicians and ancillary providers to recognize these unique adverse effects early in the course to limit organ damage. Challenges in recognition of irAEs can be attributed to their nonspecific nature, the variable spectrum of symptom severity, and timing of onset. In addition, the development of irAEs is not dose-dependent and varies among cancers treated with different drugs.

Another challenging avenue is the unpredictability of the at-risk population for developing irAEs, similar to varying spectrum of clinical response to ICI therapy; however, biomarkers have flourished, guiding clinicians. In terms of irAE susceptibility, authors have revealed that host genetic background plays a role. Noha et al. have identified eight genes that led to polymorphisms associated with an increased risk of irAEs [5]. More research is underway on this topic and will help identify patients with a high likelihood of developing irAEs, thereby allowing clinicians to maintain heightened surveillance.

Immunohistochemistry has been used to predict response to treatment (PD-L1 staining), and similarly, tumor mutational burden (TMB has been implicated as a biomarker for extended therapy response [6]. Likewise, there is a recent interest in biomarkers that can predict irAEs. Bomze et al. have concluded that high TMB can be a valuable biomarker for assessing irAE risk during anti-PD-1 therapy [7]. This is relevant to tumors with high TMB, such as melanoma and NSCLC.

### 1.1. Timeline of irAEs in Malignancies

Weber et al. have described the timeline of irAEs related to ipilimumab therapy. Dermatological and gastrointestinal manifestations were seen earlier (3–5 weeks) and resolved by week 10. Endocrinopathies and hepatotoxicity occurred later, typically between weeks 6 and 8 [8]. In general, pneumonitis has been seen later in the course (weeks 10–12) but can occur earlier with NSCLC (week 8) [9,10,11]. They can occur anytime during the treatment course and rarely even after treatment completion [12]. Delayed immune-related events (DIREs) (>90 days after treatment discontinuation) occur likely due to the lasting presence of ICIs at receptor site (receptor occupancy) despite short serum half-life (12–20 days following a single dose of nivolumab), which also explains their long-lasting activity despite treatment discontinuation [13]. DIREs call for extended pharmacovigilance post-treatment [14]. In our review, we also attempt to explain ICI rechallenge and multiorgan irAEs, which is of particular interest because information regarding these types of irAEs is currently sparse [15].

### 1.2. Overlay and Incidence of irAEs in Non-Small-Cell Lung Cancer (NSCLC)

Although any organ system can be affected, immune-related adverse events most commonly involve the gastrointestinal tract, endocrine glands, skin, and liver [9,13,14]. In a meta-analysis by Peng-Fei Wang et al., the overall incidence of irAEs in all malignancies was 26.82% for any grade and 6.10% for severe grade, while the incidence of death due to irAEs was around 0.17% [16]. A large systematic review analysis in all malignancies involving 48 trials and 6938 patients characterized patterns of irAEs based on the type of ICI. Grade 3–4 irAEs were more common with anti-CTLA-4 agents when compared with anti-PD-1 (31% vs. 10%). While colitis, hypophysitis, and rash were common with anti-CTLA-4 ICIs, pneumonitis, hypothyroidism, arthralgias, and vitiligo were common with anti-PD-1 therapy. In NSCLC, a systematic analysis involving 23 studies evaluated patients who received ICIs; the overall incidence of irAEs was 64% with PD-1 and 66% with PD-L1 inhibitors. Toxicities ≥ grade 3 were 14% and 21%, respectively, with PD-1 and PD-L1 inhibitors [17]. A phase 1 study evaluated the toxicity profile of nivolumab in NSCLC patients and found 40% to have irAEs of any grade, predominantly low grade [13]. Different types of irAEs seen in different malignancies when treated with the same class of ICIs are also of interest. For instance, higher rates of pneumonitis have been seen in NSCLC patients treated with anti-PD-1 than in melanoma patients [18].

The most common irAEs in NSCLC associated with nivolumab are rash and diarrhea, and those associated with pembrolizumab are thyroid toxicities. The incidence of irAEs with pembrolizumab was higher than with nivolumab [16]. Owen and colleagues showed that approximately 30% of patients with NSCLC treated with ICIs develop irAEs [19].

### 1.3. Contraindications to Immunotherapy

There are limited data on contraindications to immunotherapy. Patients with known autoimmune disorders have been excluded from clinical trials due to concerns regarding disease exacerbation. In addition, candidates with chronic corticosteroid use and suppressed immune systems are not thought to be candidates for immunotherapy. However, recent observational studies suggest that ICI therapy can be used safely in patients with autoimmune conditions [20]. These patients may be at an increased risk of specific irAEs (for instance, a flare-up of immune-mediated diarrhea and colitis in patients with preexisting inflammatory bowel disease [21]). Influenza vaccine administration in patients receiving ICI therapy was not associated with increased irAEs [22].

### 1.4. Principles of Generalized Management of irAEs

Management varies based on the severity of irAEs, as explained in detail below. Many societies, such as the National Comprehensive Cancer Network (NCCN), American Society of Clinical Oncology (ASCO), European Society for Medical Oncology (ESMO), and Society for Immunotherapy of Cancer (SITC), provide guidelines to manage side effects associated with ICI treatments. We summarized the incidence and management of irAEs with a focus geared towards NSCLC. Grading of irAEs is based on severity according to National Cancer Institute’s Common Terminology Criteria for Adverse Events version 5.0 (CTCAE): grade 1: asymptomatic, minimally symptomatic, or radiographic or laboratory change; grade 2: mild to moderate or persistent symptoms; grade 3: moderate or persistent symptoms; grade 4: life-threatening symptoms. Pretreatment evaluation and regular monitoring are necessary for prompt diagnosis (Table 1).

Pretherapy assessment that would be required in clinical practice for patients undergoing immunotherapy is explained in the table (Table 1).

The general principles of management of irAEs are as follows [23,24]:Corticosteroids are the mainstay of treatment for most irAEs.For grade 1 toxicities, continue ICI therapy with close monitoring except for certain hematologic, cardiac, and neurologic toxicities.For grade 2 toxicities, consider suspending ICI therapy, and treatment can be resumed when symptoms revert to grade 1 or less. Administer corticosteroids if symptoms persist more than one weekFor grade 3 toxicities, suspend ICIs and start high-dose corticosteroids, prednisone, or methylprednisolone. Taper corticosteroids over 4 to 6 weeks and maybe longer, about 6–8 weeks, particularly in irAEs such as pneumonitis or hepatitis. Refractory cases with no response in 48–72 h may require immunosuppressive therapy with agents such as infliximab. This can be repeated at weeks 2 and 6 weeks if symptoms persist. Additionally, consultation of appropriate specialists in refractory cases is encouraged. When the toxicities drop to grade 1, consider rechallenging but with caution, especially in those early-onset toxicities.For grade 4 toxicities, ICIs should be discontinued permanently, except for endocrinopathies controlled with hormone replacement.Consider prophylaxis against fungal and other microbes.Prophylaxis with proton pump inhibitors or histamine H2 blockers for gastritis, vitamin D and calcium supplementation for bone loss (osteopenia, osteoporosis), and TMP-SMX for pneumocystis pneumonia should be considered.Test for hepatitis B and C and latent/active tuberculosis due to the risk of reactivation before initiating TNF inhibition (e.g., infliximab).Consider vedolizumab, an alpha-4 beta-7 integrin inhibitor, for immune-related hepatitis. In irAE-related hepatitis, avoid infliximab.

## 2. Organ-Specific Immune-Related Adverse Events

### 2.1. Fatigue

Fatigue is the most common symptom reported by up to 40% of patients after treatment with anti-CTLA-4 antibodies [25] and in 16–24% of patients treated with anti-PD-1/anti-PD-L1 antibodies seen in single-agent trials. A similar incidence of 19–21% was observed in a systematic review involving 5744 NSCLSC patients treated with PD-1 and PD-L1 inhibitors [17]. Fatigue is usually mild; if severe, it should trigger an assessment for underlying disorders such as endocrinopathies [26]. It is graded as mild, moderate, or severe. Grade 1 often resolves with rest and does not affect activities of daily living (ADLs); immunotherapy can be continued in such cases. Grade 2, moderate fatigue, is usually not relieved by rest and limits ADLs. Immunotherapy can be continued with supportive care for symptoms; low-dose steroids can be considered. In grade 3, severe fatigue limiting self-care (ADLs), consider discontinuing immunotherapy based on factors such as disease progression, medical comorbidities, and presence of other irAEs, if any. Management of fatigue includes clinical exam; medication review; routine laboratory exam; and hormone testing for thyroid (TSH, free T3 and T4), morning cortisol, morning ACTH if low cortisol, and morning testosterone levels (recommended for men). Management of fatigue includes educating the patient on sleep, diet, hydration, and medications.

### 2.2. Infusion-Related Reactions (IRRs)

Infusion reactions, including fever and chills, are more familiar with CTLA-4 inhibitors, accounting for adverse effects in phase 3 studies [27]. Mild infusion reactions have been reported in up to 25% of patients with anti-PD-1 or anti-PD-L1 agents; high-grade, severe, and life-threatening events have been reported in <2% [28]. The majority occurred during the first infusion, with nearly all reactions occurring within the first four treatment cycles. The majority of these can be managed supportively with antipyretics and antihistamines. Premedication appeared to decrease the rate of severe IRRs [29]. Common symptoms include fever, chills, rigors, urticaria, pruritis, angioedema, flushing, headache, nausea, hypertension, hypotension, dyspnea, cough, and wheezing. Other clinical features such as hypoxemia, dizziness, syncope, sweating, arthralgias, and myalgias may also occur. Management includes a thorough physical examination, monitoring vital signs including pulse oximetry, and obtaining an ECG for chest pain or sustained tachycardia. Grades of severity are based on duration of symptoms and response to symptomatic treatment. Grade 1 is transient; ICI can be continued with close monitoring. Alternatively, treatment can be held until symptom resolution. Grade 2 symptoms respond to conservative management ideally in less than 24 h. ICI therapy can be continued, but consider slowing the rate of infusion while premedicating with future infusions. Grades 3 and 4 have prolonged symptoms and do not respond immediately to conservative management. Recurrence can occur following initial improvement. ICI therapy should be permanently discontinued in case of recurrence. Severe cases might require hospitalization to prevent life-threatening complications.

### 2.3. Dermatological Manifestations

Skin manifestations, such as rash/pruritus and mucositis, are the most common irAEs associated with ICIs. Skin toxicity is an early irAE ranging from rash or pruritus to vitiligo. The characteristic rash is faintly erythematous and maculopapular, involving the trunk and extremities, and may be pruritic [30]. Vitiligo can also occur commonly and has a delayed appearance after several months of treatment [31]. Approximately 47–68% of patients treated with anti-CTLA-4 antibodies and 30–40% of patients treated with anti-PD-1/anti-PD-L1 antibodies suffer skin toxicities of any grade [32]. Rash and pruritus are reported in up to 10% of patients treated with anti-PD-1 inhibitors, with incidence increasing to 15% and 25% with the anti-PD-1 and anti-CTLA-4 or chemotherapy combination [33]. Grades 1 and 2 are defined as macules/papules covering <10% and 10–30% of the body surface area, respectively, while grade 3 involves >30% of BSA.

Most skin eruptions are mild, and immunotherapy can be continued in most patients [32]. Management should be focused on ruling out any other skin toxicity etiologies, such as an infection, drug interaction, or another systemic disease. A complete physical examination is essential, including mucosal areas for defining the body surface area (BSA) affected by the toxicity and biochemical evaluation of liver and kidney function. The treatment of symptomatic cases with a topical glucocorticoid such as betamethasone 0.1% cream, urea-containing cream, or an oral antipruritic agent (diphenhydramine, hydroxyzine, GABA agonists, or NK-1 receptor antagonists) is usually sufficient [32,34]. Grade 3 cases should be treated with oral glucocorticoids for 3–4 weeks, with temporary discontinuation of ICIs. Consider permanent discontinuation in more severe cases such as Stevens–Johnson syndrome, but fortunately, severe irAEs are rare. Patients who fail to respond to steroids or exhibit the formation of bullae merit dermatologic evaluation and skin biopsy [9,26,32,33].

### 2.4. Musculoskeletal Toxicities

The common musculoskeletal toxicities with immune checkpoint inhibitors are inflammatory arthritis, myalgias/myositis, polymyalgia rheumatica, and giant cell arteritis. The prevalence and grading are different for each of the above presentations, as outlined below [9,22,32,34,35].

#### 2.4.1. Inflammatory Arthritis

Inflammatory arthritis is defined as inflammation of the joints with a prevalence of 5–12%, depending on the type of ICI used. The lowest prevalence was with ipilimumab, and the highest prevalence was with pembrolizumab. The number of joints involved, functional assessment, and imaging (X-rays, joint ultrasound, or MRI) help with grading. Mild irAEs can be controlled with NSAIDs. If NSAIDs are ineffective, low-dose prednisone at 10–20 mg daily for 4 weeks can be considered, while intra-articular steroids can be considered depending on the joint location and number involved. Grade 2 is moderate pain with inflammation, erythema, and joint swelling limiting ADLs. Consider holding ICIs and starting prednisone at 0.5 mg/kg/day for 4–6 weeks, and if there is no improvement by 4 weeks, consultation with rheumatology is strongly recommended. Grades 3 and 4 involve severe pain associated with the above signs, disabling/limiting self-care with activities of daily living. In these patients, hold or permanently discontinue ICI and start high-dose prednisone/methylprednisolone 1 mg/kg; if no improvement is seen by 2 weeks, consider rheumatology consultation and adding DMARDs.

#### 2.4.2. Myositis

Myalgias or myositis presents as muscle weakness and is the second most common musculoskeletal irAE, with a prevalence range of 2–12%. Laboratory tests include CMP, CK, aldolase, and inflammatory markers such as ESR and CRP. Consider EMG, imaging with ultrasound/MRI, or biopsy on an individual basis. When the diagnosis is uncertain, muscle strength testing and paraneoplastic autoantibody testing for myositis and neurological conditions such as myasthenia gravis should be considered. Grade 1 is mild weakness with or without pain. Consider holding ICI, monitor serial aldolase and CK levels to treat pain with NSAIDs, and consider ruling out polymyalgia rheumatica/giant cell arteritis. Grade 2 is moderate weakness with or without pain, limiting age-appropriate instrumental ADL. Consider holding ICI and monitor serial aldolase and CK levels to treat pain with NSAIDs. Consider prednisone dose < 10 mg, as well as rheumatology consultation. For CK levels > 3 times upper limit of normal, initiate prednisone or equivalent at 0.5–1 mg/kg; treat as grade 3 if symptoms worsen. Grades 3 and 4 are severe weakness with or without pain, limiting self-care ADLs. Hold ICI; consult rheumatologist/neurologist; and consider MRI, EMG, and muscle biopsy in severe or refractory cases. Monitor aldolase and CK until symptoms have resolved. In addition, consider IVIG (2 g/kg administered in divided doses). Plasmapheresis, infliximab, or mycophenolate may be considered if myositis is refractory to steroids.

#### 2.4.3. Polymyalgia Rheumatica or Giant Cell Arteritis

Polymyalgia rheumatica (PMR) or giant cell arteritis has been reported in case reports and case series, and the exact prevalence is unknown. Consider ESR, CRP, ANA, RF, anti-CCP, and temporal artery ultrasound and biopsy if visual symptoms/headache are present. Consider ultrasound of shoulders or hips if PMR is suspected. Grade 1 is mild stiffness and pain. Continue ICI and consider acetaminophen/NSAIDs for pain. Grade 2 is moderate stiffness and pain, limiting age-appropriate instrumental ADLs. Consider holding ICI, initiate symptomatic treatment in addition to prednisone 10 mg with a taper over 4–6 weeks, and obtain rheumatology consultation. Treat as grade 3 if symptoms worsen. Grades 3 and 4 are severe stiffness and pain, limiting self-care ADLs. Hold ICI, resume if symptoms drop to grade 1, start prednisone 10–20 mg/day and taper over 8–12 weeks, and have rheumatology consultation. If giant cell arteritis is suspected, hold immunotherapy, start prednisone 1 mg/kg/day, and taper over 8–12 weeks. If visual symptoms are present, consider pulse-dose steroids (methylprednisone at 500–1000 mg/day), ophthalmology, and rheumatology consultation.

### 2.5. Gastrointestinal Toxicity

The gastrointestinal tract is one of the most common systems involved in immunotherapy-related adverse events, with immune-mediated diarrhea and colitis (IMDC) being the most common manifestation, followed by nausea [35]. Diarrhea and colitis are part of the same clinical spectrum, with diarrhea being defined as increased frequency of stools and colitis being associated with abdominal pain in addition to evidence of inflammation, either radiographic or endoscopic. It most commonly occurs 4 to 6 weeks into therapy [36]. The incidence of diarrhea is higher in patients receiving CTLA-4 blocking antibodies than PD-1 inhibitors [37]. The overall incidence of colitis in NSCLC treated with combination ICI therapy is 20–32%, with about 11% of grade 3–4 severity [38,39,40]. In patients receiving PD-L1 inhibitors in advanced nonsquamous non-small-cell lung cancer, the incidence rate of diarrhea was observed to be 8% [41]. Increased risk of IMDC has been seen in patients with pre-existing inflammatory bowel disease [42].

Grading is based on the frequency of stools and the occurrence of complications. Grade 1 is defined as increased stool frequency up to four times/day above baseline. Immunotherapy can be continued in such patients, but alternatively, holding immunotherapy until toxicity resolves is also appropriate. It is imperative to take a detailed history (exposure to foods, drugs, freshwater, travel, etc.) and rule out infectious etiologies such as *Clostridioides* (formerly *Clostridium*) *difficile* and other viral or bacterial infections [43]. Management of all patients includes diagnostic workup with CBC, BMP, LFT, standard microbiological workup including stool culture, stool for ova and parasites, and testing for infection. Based on institutional availability, consider testing for stool lactoferrin or calprotectin. If positive for these tests, strongly consider esophagogastroduodenoscopy (EGD) or flexible sigmoidoscopy with biopsy within the first 2weeks of the onset of symptoms [44]. Antimotility drugs such as loperamide or atropine/diphenoxylate can be used in grade 1 symptoms once infectious etiologies have been ruled out. Monitor for dehydration and consider dietary modifications such as BRAT or FODMAP. If the symptoms occur for more than a few days, check lactoferrin; if negative, continue antimotility management with likely addition of mesalamine or cholestyramine. However, if lactoferrin is positive, treat it as grade 2 colitis. Grade 2 colitis is defined as 4–6 stools per day above the baseline or a moderate increase in stool output from the colostomy. In addition to management as for grade 1, consider computed tomography (CT) scan of abdomen and pelvis and endoscopic imaging in the form of esophagogastroduodenoscopy, colonoscopy, or flexible sigmoidoscopy with biopsy. Check calprotectin every 2 months to monitor trends and guide treatment. Stop treatment upon return to normal/negative. Consider inpatient management and start prednisone or methylprednisolone at the dose of 1 to 2 mg/kg/day and hold immunotherapy. If there is no response in 2 to 3 days with steroids, consider infliximab or FDA-approved biosimilars after completing appropriate screening laboratory work-up testing for human immunodeficiency virus (HIV), hepatitis A, hepatitis B, and blood QuantiFERON for tuberculosis.

Grade 3 severity is defined as having more than seven stools per day above baseline or severe output from the colostomy bag. Grade 4 toxicity is the occurrence of life-threatening complications such as bowel perforation, peritonitis, ischemic necrosis, and toxic megacolon. Discontinue anti-CTLA-4 antibody therapy permanently and consider resuming anti-PD-1 or PD-L1 after the resolution of gastrointestinal toxicity. Permanently discontinue the immunotherapy agent responsible for grade 4 symptoms. If there is no improvement with oral steroids, consider inpatient management with intravenous steroids or fluids [32,43]. If there is no benefit from intravenous steroids in 2 to 3 days, consider infliximab or vedolizumab after appropriate initial screening workup with close monitoring for complications. Gastroenterology consultation is strongly recommended in the inpatient setting. The treatment paradigm in this space is evolving. More research is required regarding the duration of therapy of TNF-alpha blockers and the management of colitis refractory to TNF-alpha blockers. Evidence supports up to three doses of TNF-alpha blockers with reduced recurrence rates. Consider a repeat colonoscopy with biopsy if symptoms recur or progress to grade 3 or 4. However, repeating colonoscopy to prove the clearance of colitis is optional. Complications such as perforation require acute surgical intervention. Colitis treatment should continue until the symptoms improve to grade 1 or less, and steroids should be tapered over 4 to 6 weeks [45,46].

### 2.6. Hepatotoxicity

Hepatotoxicity predominantly manifests as elevated liver enzymes with hepatocellular pattern and occurs approximately 8 to 12 weeks after initiation of treatment [47]. However, median time to onset of 1.9 weeks was noted in NSCLC, with resolution in the majority of patients within 8 weeks of appropriate treatment [48]. Most patients are asymptomatic; abnormality is incidentally noted on laboratory evaluation and rarely associated with fever [49].

Like in IMDC, the incidence of hepatotoxicity of any- and high-grade is higher with CTLA-4 inhibitors when compared to PD-1 inhibitors [50]. In NSCLC treated with PD-1/PD-L1 inhibitors, the incidence is 2–7.6% and ~1% for all-grade and high-grade (≥3), respectively [35,48]. Higher all- and high-grade hepatotoxicity has been seen with first-line nivolumab, while interestingly, this was not the case with pembrolizumab [51,52]. Increased incidence has been observed with combination strategies; PD-(L)1 plus CTLA-4 inhibitors or chemotherapy had incidence rates of 3–20% for all-grade and up to 9% for grade ≥ 3 toxicities [38,39,40]. Severity is graded based on the level of elevation of AST/ALT and bilirubin levels. It is recommended for all patients to receive a baseline liver function test before immunotherapy initiation and prior to every treatment. Monitor patients with regular liver function tests (LFTs) based on severity.

Grade 1 hepatotoxicity is defined as an elevation of AST or ALT but <3 times the upper limit of normal (ULN). All patients with hepatotoxicity should be tested for acute viral hepatitis. Limiting or discontinuing hepatotoxic medication should be considered. Continue immunotherapy for grade 1 severity with weekly monitoring of LFTs. Grade 2 hepatotoxicity is defined as an elevation of AST or ALT up to 3–5 times the ULN. Hold immunotherapy for these patients; monitor LFTs at an increased frequency of every 3 to 5 days. Start oral prednisone at a dose of 1 mg/kg/day. Consider liver ultrasound. If unremarkable, consider magnetic resonance cholangiopancreatography (MRCP) of the abdomen. Typically imaging changes are not seen with abnormal LFTs; occasionally, imaging can reveal hepatomegaly, periportal edema, or periportal lymphadenopathy [53].

Grade 3 is defined as an elevation of AST/ALT up to 5–20 times ULN, and grade 4 includes life-threatening complications and elevation of AST and ALT above 20 times ULN. Immunotherapy should be held for grades 3 and 4. Patients with grade 2 with bilirubin elevation more than 1.5 times ULN should receive severe grade management. Intravenous steroids at a dose of 1 to 2 mg/kg/day are recommended with monitoring of LFTs and opinion from a liver or gastroenterology specialist. Consider adding mycophenolate mofetil 0.5 to 1 g every 12 h if steroid-refractory after 2 to 3 days. Due to potential hepatotoxicity, infliximab is contraindicated. Once sustained improvement of liver function tests to levels less than grade 1 is attained, steroid tapering is indicated, tapering over at least 4–6 weeks. For patients with unclear liver function abnormalities, consider a liver biopsy; biopsies occasionally show panlobular hepatitis with prominent perivenular infiltrate with endotheliitis [45,54]. A study by De Martin et al. characterized distinct histological profiles based on ICI type: granulomatous hepatitis including fibrin ring granulomas and central vein endotheliitis with CTLA-4 antibodies and lobular hepatitis with PD-(L)1 antibodies. In addition, recent studies suggest the need for patient-tailored management as spontaneous resolution has been observed in about 35–50% of patients [55,56].

### 2.7. Endocrine Toxicity

#### 2.7.1. Incidence

ICI-related endocrine gland autoimmunity has resulted in the dysfunction of the thyroid, pituitary, and adrenal glands and pancreas. Manifestations of immune-mediated endocrine gland dysfunction include hypothyroidism, hyperthyroidism, hypophysitis, type 1 diabetes, and primary adrenal insufficiency. Overall, combination ICI therapy was associated with the highest incidence of endocrinopathy [57,58]. The median time to onset of moderate to severe endocrinopathy ranges between 1.75 and 5 months for ipilimumab. The median time to onset of endocrinopathy with PD-1 inhibitor monotherapy ranged from 1.4 to 4.9 months [59,60]. The estimated incidence of hypothyroidism was 3.8% with ipilimumab and up to 13.2% for combination therapy. Overall, the observed incidence of hypophysitis was 6.4% for combination therapy, 3.2% for CTLA-4 inhibitors, 0.4% for PD-1 inhibitors, and less than 0.1% for PD-L1 inhibitors [61,62]. All patients who developed or experienced worsening of diabetes (i.e., becoming insulin-dependent) had received anti-PD-1/PD-L1 therapy. The median time to onset was 20 weeks after the first ICI cycle; 59% presented with ketoacidosis, 42% had evidence of pancreatitis, and 40% had one or more positive autoantibodies on testing [59,61,63,64]. About 1–10% incidence of immune-related endocrinopathies in NSCLC has been seen with anti-PD-1 and PD-L1 inhibitors, while combination therapy including CTLA-4/PD-(L)1 antibodies was noted to have an incidence of 11–23% [63].

#### 2.7.2. Diabetes

Fasting glucose level is preferred to assess potential hyperglycemia. Management is guided by the history of type 2 diabetes mellitus (T2DM), glucose levels, and concern for diabetic ketoacidosis (DKA). For patients with new-onset hyperglycemia less than 200 mg/dL and a history of T2DM with low suspicion for DKA, ICIs can be continued with serial blood glucose monitoring at each dose. Further workup is warranted for findings of (1) new-onset hyperglycemia > 200 mg/dL, (2) random blood glucose > 250 mg/dL, or (3) history of T2DM with glucose > 250 mg/dL. If any of these findings are noted, consider new-onset type 1 diabetes mellitus (T1DM) and evaluate for DKA. ICI-related development of T1DM is rare (1–2%) and can be life-threatening, and insulin therapy is essential in these patients. Endocrinology input is quite important in managing such cases; monitoring DKA requires hospitalization and a temporary hold of ICI therapy. Tight glycemic control can have improved cardiovascular outcomes, particularly in critically ill patients, including acute coronary patients [64].

#### 2.7.3. Thyroid Dysfunction

Thyroid dysfunction is the most common endocrine disorder in NSCLC with a preponderance towards hypothyroidism (incidence of ~10% with combination strategies), while hyperthyroidism and thyroiditis are less common [35]. Thyroid function should be assessed by monitoring thyroid-stimulating hormone levels (TSH) and free thyroxine (T4). In the setting of thyroid abnormalities, routine monitoring is recommended every 4 to 6 weeks. For asymptomatic or subclinical hypothyroidism (elevated TSH with normal free T4), continue regular monitoring while continuing ICIs. Consider levothyroxine for TSH levels above 10 mIU/L. Primary hypothyroidism is characterized by high TSH levels (>10 mIU/L) and low free T4 with clinical symptoms. Provide thyroid supplementation and consider endocrine consultation. Before starting thyroid replacement therapy, concomitant adrenal insufficiency should be ruled out by testing AM cortisol levels. Low or suppressed TSH with inappropriately low free T4 may present as a sequela of hypophysitis, in which other pituitary axes may be affected. Follow free T4 for thyroid replacement in the setting of hypophysitis-induced loss of TSH production. Although rare, thyroiditis (often a painless, transient inflammatory process) can occur with ICI. Thyrotoxicosis often evolves to hypothyroidism. Findings of persistently suppressed TSH with high free T4/total T3 require additional testing to rule out true hyperthyroidism and Graves’ disease-like etiology [65].

#### 2.7.4. Hypophysitis

The incidence of hypophysitis in NSCLC patients treated with anti-PD-1/PD-L1 is less than 1%, with no increase with ICI combination therapy. Acute hypophysitis symptoms include headache, photophobia, dizziness, nausea/emesis, fevers, anorexia, visual field cuts, and severe fatigue. Chronic symptoms can include fatigue and weight loss. Workup for hypophysitis includes assessing ACTH, morning cortisol, FSH, LH, TSH, free T4, testosterone in men, and estrogen in premenopausal women. Test results indicative of hypophysitis may show low levels of ACTH, cortisol, sodium, potassium, testosterone, and DHEA-S. If the patient is symptomatic, it is crucial to obtain an MRI brain with pituitary/sellar cuts. For acute, symptomatic hypophysitis (headache and symptoms caused by acute swelling of the pituitary), hold immunotherapy and initiate methylprednisolone/prednisone at 1–2 mg/kg/day until acute symptoms resolve, typically 1 to 2 weeks. Then taper steroids rapidly to physiologic replacement levels upon improvement. Consider resumption of ICI therapy once symptoms related to mass effect have resolved. The more common presentation for hypophysitis features deficiency of pituitary hormones TSH and ACTH and gonad-stimulating hormones but without symptomatic pituitary swelling. Immunotherapy can be continued while endocrine therapy is titrated to appropriate physiologic levels [66].

Workup for primary adrenal insufficiency should include serum cortisol, as well as CMP and renin levels. Hallmarks of adrenal damage include low morning cortisol (<5) with ACTH above the reference range, with or without abnormal electrolytes and symptoms. Endocrinology should be consulted for these patients, with specialist evaluation before any surgery or procedure. Hold immunotherapy. If patients are hemodynamically unstable, recommend inpatient care and high-dose/stress-dose corticosteroids. Immunotherapy can be resumed once endocrine replacement therapy has been established. Physiologic hormone replacement will likely be required indefinitely (typically life-long) [57,58,60].

### 2.8. Pulmonary Toxicity

Pneumonitis is a relatively uncommon but fatal complication associated with ICIs. For PD-1/PD-L1 monotherapy, rates of any-grade pneumonitis are at or below 5%, and the rate of high-grade pneumonitis is around 1% [9]. Ipilimumab monotherapy has a lower incidence of pneumonitis than PD-1/PD-L1 inhibitors, with reported rates of less than 1% [33,67]. The median time to irAE onset from the start of treatment is 2.5 months, with earlier onset for the combination versus monotherapy [9,68]. The overall incidence of any-grade and high-grade pneumonitis is 2.7% and 0.8%, respectively, with a higher incidence in NSCLC than melanoma. In addition, when compared to melanoma (5.2 months) and other cancers, the time to onset of irAEs is shorter in NSCLC (2.1 months) [10,11]. Incidence of pneumonitis for PD-1/PD-L1 inhibitor monotherapy versus combination therapy with CTLA-4 inhibitor was 3% and 10%, respectively [5,6,48]. Cases of radiation recall pneumonitis involving previously irradiated lung fields have been reported, and this is of particular interest in lung cancer patients [69].

Mild pneumonitis (grade 1) is usually asymptomatic and confined to less than 25% of the lung parenchyma or a single lobe. Moderate pneumonitis (grade 2) is characterized by new or worsening symptoms, including shortness of breath, cough, chest pain, and fever. Severe pneumonitis (grade 3) involves all lobes of the lung or greater than 50% of the lung parenchyma. The symptoms typically limit self-care ADLs. Life-threatening (grade 4) pneumonitis involves serious respiratory compromise, requiring hospitalization and high-flow oxygen or intubation. Most low-grade pneumonitis patients can be treated in the outpatient setting, but a small subset and all patients ≥ grade 3 require inpatient care. It is crucial to rule out disease progression and infection in NSCLC. Radiological patterns and clinical history can provide clues, and if uncertain, bronchoscopy with bronchoalveolar lavage or lung biopsy can be considered. Grade 1 cases can be managed by holding immunotherapy and initiating oral corticosteroids, while all moderate and severe cases require systemic corticosteroids. Among patients with grade 3 or higher pneumonitis, additional immunosuppression with infliximab alone or infliximab with cyclophosphamide might be indicated.

### 2.9. Renal Toxicity

The estimated incidence of all-grade renal toxicity is approximately 2% for NSCLC patients treated with nivolumab or pembrolizumab monotherapy; the incidence of high-grade renal toxicity was 0.6% [52,70,71]. The incidence of grade 1–2 irAEs reaches 7.5% for ICI combination therapy with CTLA-4 inhibitors such as ipilimumab with PD-1/PD-L1 inhibitors and up to 4% for grades 3–4 [48,72]. The time to onset of renal toxicity has been reported to be around 6 to 12 weeks for ipilimumab and 3 to 12 months for PD-1 inhibitors [73]. About half of the renal irAEs resolve without the use of corticosteroids or immunosuppression [70].

Elevated serum creatinine could indicate developing renal irAEs. Signs of acute renal failure may include azotemia, creatinine elevation, inability to maintain acid/base or electrolyte balance, and reduced urine output. Mild renal irAEs (grade 1) are categorized by serum creatinine level 1.5 to 2 times above baseline or an increase of ≥0.3 mg/dL. Creatinine levels of 2 to 3 times above baseline are considered moderate renal irAEs (grade2). With severe irAEs (grade 3), creatinine levels may be more than three times above baseline or >4.0 mg/dL. Creatinine levels > 6 times above baseline indicate life-threatening renal issues (grade 4) and necessitate dialysis. For mild to moderate renal irAEs, follow creatinine and urine protein every 3 to 7 days. Consider holding immunotherapy for grade 1 renal dysfunction, and hold immunotherapy for grade 2 irAEs. After improvement to ≤grade 1, consider resuming immunotherapy with concomitant corticosteroids. Permanently discontinue immunotherapy if severe/life-threatening renal irAEs occur. Consider inpatient care, nephrology consultation, renal biopsy, and initiating methylprednisone/prednisolone at 1–2 mg/kg/day [70,74,75].

### 2.10. Ocular Toxicity

The affected area of the eye categorizes ophthalmic irAEs into ocular inflammation (e.g., uveitis, episcleritis, blepharitis, peripheral ulcerative keratitis), orbital inflammation/orbitopathy (e.g., idiopathic or thyroid-induced orbitopathy), retinal/choroidal disease (e.g., retinopathy or choroidal neovascularization), and optic neuropathy [71,76]. Dry eye and uveitis have been the most commonly reported ocular ICI-associated events, with a reported incidence between 1% and 24% [74,77]. The incidence of irAEs in NSCLC treated patients is unknown. Mild ophthalmic irAEs have generally been managed successfully using artificial tears and topical steroids, whereas more severe conditions have required systemic corticosteroid therapy and discontinuation of ICI therapy. For grade 2 or higher toxicity, immunotherapy is held, and urgent ophthalmology consultation is indicated [71,76,77].

### 2.11. Neurological Toxicity

Neurologic manifestations are among the less common immune-related adverse events [78]. The presentation spectrum varies, with the most common being headache and peripheral neuropathy [75]. Other reported severe neurologic complications include myasthenia gravis, posterior reversible encephalopathy syndrome (PRES), Guillain–Barré syndrome (GBS), aseptic meningitis, autonomic neuropathy, transverse myelitis, autoimmune encephalitis, and cranial and peripheral neuropathies. Symptoms occur within the first 3 months of the immunotherapy initiation and are observed to improve in about 75% of the cases after discontinuation of therapy [78,79]. The incidence varies from 1% to 14% and is around 4% for CTLA-4 blockade, 6% for PD-1 inhibitors, and 12% with combined CTLA-4 and PD-1 inhibition [24,80]. The ratio of the incidence of peripheral irAEs to central irAEs is higher in anti-PD-1 cases (2.2:1) than in anti-CTLA-4 cases (1.7:1) [35]. The evolution of neurological irAEs can be rapid, unpredictable, and life-threatening [80]. The time to onset was from 6 to 12 weeks, but some delayed presentations can occur 6–8 months out [28]. Among NSCLC patients, there is no reported incidence of neurological irAEs, but they have been reported in a few SCLC cases [81,82,83]. Given broadly varying clinical spectrum and nonspecific presentation, neurological irAEs are particularly difficult to diagnose, posing a challenge to oncologists [79,84]. Individual toxicities and their management are beyond the scope of this paper.

The severity of presentation is graded based on their effect on the activities of daily living. Grade 1 is asymptomatic or mildly symptomatic and does not interfere with life activities. Grade 2 is a moderate degree of symptoms, limiting instrumental activities of daily living (IADLs). Grade 3 or 4 is a severe grade with symptoms affecting self-care and ADLs, with patients relying on aids for self-care. Grade 4 comprises life-threatening symptoms. Close coordination with neurology specialists is critical for prompt diagnosis and optimal treatment [78,79].

In peripheral and autonomic neuropathy, it is important to screen for reversible causes such as diabetes, vitamin deficiencies (B12, folate), TSH, HIV, serum protein electrophoresis, vasculitis, and autoimmune disorders. Inflammatory markers such as ESR, CRP, CPK, aldolase, and anti-striated antibodies are helpful in the event of superimposed myositis. Neurology consultation is necessary. Other modalities such as MRI brain or spine, lumbar puncture for CSF analysis, and nerve conduction studies depending upon the symptoms are needed to rule out other neurological disorders. For peripheral and autonomic neuropathy, hold immunotherapy and restart if severity returns to grade 1. However, in case of transverse myelitis of any grade and grade 2 myasthenia gravis, GBS, encephalitis, and aseptic meningitis, consider permanent discontinuation of immunotherapy. Steroids are again the mainstay of treatment for neurological irAEs in addition to other sorts of immunosuppression such as IVIG and plasmapheresis in certain situations. Given the complexity in clinical presentation and difficulty in diagnosis, neurology consultation is vital, and patients need admission to the hospital for acute presentations and emergent treatment [28,80,81,83,84].

### 2.12. Cardiovascular Toxicity

Various cardiovascular irAEs include pericarditis, myocarditis, pericardial effusions, nonischemic cardiomyopathy, and conduction abnormalities. Epidemiological data are sparse for other rare effects, including vasculitis, destabilized atherosclerotic lesions, conduction abnormalities, venous thromboembolism, and takotsubo syndrome [85]. The most common irAE associated with cardiac toxicity is myocarditis, with a prevalence ranging from 0.06% to 2.4% and higher with combination immunotherapy with a 27–46% mortality rate [85,86]. Very few case series and case reports have been published in the literature about cardiac events in NSCLC patients receiving ICIs due to the rarity of the cardiac side effects compared to other immune-related events, making it difficult to estimate the prevalence [86]. The incidence of cardiac irAEs in NSCLC patients in clinical trials was found to be up to 0.15% [87,88]. Pleural and pericardial effusions have been noted more frequently in patients with malignant pleural involvement [89,90].

The symptoms of myocarditis are usually nonspecific. Patients need to be monitored on telemetry due to the high risk of mortality, and an urgent cardiology consultation is indicated. Electrocardiogram, cardiac biomarkers (CPK, CK-MB, and troponin), and inflammatory biomarkers need to be obtained. Viral titers to evaluate for other causes, echocardiogram, and cardiac MRI may be considered for extensive evaluation, including a cardiac biopsy at times. Grade 1 cardiotoxicity is asymptomatic with abnormalities in laboratory studies or cardiac imaging. Grade 2 symptoms occur with moderate activity, while grade 3 symptoms occur at rest or with minimal activity or exertion. Grade 4 involves life-threatening consequences. If the patient experiences grade 3–4 symptoms, immunotherapy is permanently discontinued. Close ICU monitoring is recommended. Consider methylprednisolone pulse dosing at 1 g/day for 3–5 days until the cardiac function returns to baseline, then taper over 4–6 weeks; if no improvement is seen in 24 h on steroids, consider adding other immunosuppressive agents such as anti-thymocyte globulin (ATG), infliximab, IVIG, and mycophenolate. A temporary pacemaker might be required in patients with life-threatening arrhythmias [85,86].

## 3. Multiorgan irAEs

Multiorgan irAEs are characterized by a combination of irAEs involving several organ systems [15]. Previous studies have reported good tumor response in patients with multiorgan irAEs. For example, a large meta-analysis involving 4324 patients has shown that the development of irAEs was associated with reduced risk of progression and reduced risk of death [91]. This is likely related to strong T-cell activation and better response to immunotherapy manifested as adverse effects. Another study by Shankar et al. described a similar finding in NSCLC patients treated with ICIs [92].

Given their correlation with tumor response, there is a need for consistent and standardized reporting of the incidence of multiorgan irAEs. Further research is needed to predict which patients might develop multiorgan irAEs and are likely to respond better to ICI therapy. It is crucial to manage these adverse effects better to obtain the maximal benefit of ICI therapy.

## 4. Immune Checkpoint Inhibitor Therapy Rechallenge

As the name suggests, rechallenge is offering ICI therapy to the patient once irAEs have subsided. There seems to be much hesitancy in rechallenge due to fear of recurrence of irAEs. A multidisciplinary approach, careful balancing of risks and benefits after discussion with patients, and increased vigilance for subsequent irAEs are essential when rechallenge is offered. Several essential elements to consider are the severity of symptoms, availability of alternative treatment options, disease progression, and the treatment response achieved at the time of discontinuation.

Caution should be exercised with careful monitoring for recurrent symptoms, especially in patients with early onset of irAEs. If moderate irAEs recur, we suggest considering discontinuation of that particular class of ICI. In severe irAEs, permanent discontinuation of that class of ICI is recommended. In addition, ICIs should be permanently discontinued in grade 4 adverse reactions except for certain endocrine disorders that can be managed with hormone replacement.

Rechallenge can be offered to a select subset of patients after careful consideration with appropriate monitoring and algorithm-based protocol to identify and treat side effects. Dolladille and colleagues assessed the safety in patients with cancer who resumed therapy with the same ICIs and found that rechallenge was associated with about 25–30% of the same irAEs experienced previously [93]. In a small group of patients with advanced NSCLC, rechallenge proved effective [94]. In a retrospective analysis by Santini et al., NSCLC patients rechallenged with anti-PD-(L)1 therapy after treatment delay due to irAE were evaluated. Eighteen out of thirty-eight patients (48%) had no subsequent irAEs. A quarter of patients each had the same irAE or a new distinct irAE. Sixty percent of the irAEs were mild (grades 1–2), and the rest were grades 3 and 4. While two retreatment deaths were reported, most events (17/20) resolved or improved to grade 1. One important conclusion is that patients requiring hospitalization for initial irAE or with complete or partial treatment response should not be rechallenged [95].

A large retrospective analysis of anti-PD-1/PD-L1 rechallenge in NSCLC by Gobbini et al. showed that only 50–55% of patients experienced irAEs of any grade. The authors also report longer median OS in patients who discontinued ICIs due to toxicity and clinical decision compared to disease progression, suggesting increased long-term survival with ICI rechallenge for reasons other than disease progression [96].

Suggested approaches for rechallenging ICIs include a class switch, resuming ICIs after complete or near resolution of irAEs, and concomitant use of immunosuppressants with ICIs. The class switch includes switching from anti-CTLA-4 to anti-PD-1 and vice versa. Anti-CTLA-4 and anti-PD-1 have slightly different mechanisms of action, enabling this switch. A retrospective study in metastatic melanoma patients demonstrated the benefits of a class switch when changed from anti-CTLA-4 to anti-PD-1 with mild adverse effects not necessitating discontinuation of therapy [97].

## Figures and Tables

**Table 1 biomedicines-10-00790-t001:** Pre-therapy assessment for patients undergoing immunotherapy.

Pretherapy Assessment	Monitoring Frequency	Evaluation for Abnormal Signs/Symptoms
Clinical: Physical examination Comprehensive patient history including autoimmune/organ-specific disease, endocrinopathy, or infectious disease Neurological examination Bowel habits (typical frequency/consistency) Infectious disease screening as indicated	Clinical exam at each visit with adverse event (AE) symptom assessment	Follow-up testing based on signs and symptoms
Imaging: Cross-sectional imaging Brain MRI if indicated	Periodic imaging as indicated	Follow-up testing as indicated based on imaging findings
General bloodwork: CBC with differential Comprehensive metabolic panel	Repeat prior to each treatment or every 4 weeks during immunotherapy, then in 6–12 weeks or as indicated	HbA1c for elevated glucose
Dermatologic: Examination of skin and mucosa if history of immune-related skin disorder	Conduct/repeat as needed based on symptoms	Monitor affected BSA and lesion type; photographic documentation. Skin biopsy if indicated
Pancreatic: Baseline testing is not required	No routine monitoring needed if asymptomatic	Serum amylase, lipase, and consider abdominal CT with contrast or MRCP for suspected pancreatitis
Thyroid: Thyroid-stimulating hormone (TSH), free thyroxine (T4)	Every 4–6 weeks during immunotherapy, then follow up every 12 weeks as indicated	Total T3 and free T4 if abnormal thyroid function suspected
Adrenal/Pituitary: Baseline testing is not required	Repeat prior to each treatment or every 4 weeks during immunotherapy, then follow up every 6–12 weeks	Luteinizing hormone (LH), follicle-stimulating hormone (FSH), testosterone (males), estradiol (females), adrenocorticotropic hormone (ACTH)
Pulmonary: Oxygen saturation (resting and with ambulation)	Repeat oxygen saturation tests based on symptoms	Chest CT with contrast to evaluate for pneumonitis; biopsy if needed to exclude other causes
Cardiovascular: Consider baseline ECG Individualized assessment in consultation with cardiology as indicated	Consider periodic testing for those with abnormal baseline or symptoms	Individualized follow-up in consultation with cardiology as indicated
Musculoskeletal: Joint examination/functional assessment as needed for patients with pre-existing disease	No routine monitoring needed if asymptomatic	Consider rheumatology referral. Depending on clinical situation, consider C-reactive protein (CRP), erythrocyte sedimentation rate (ESR), or creatinine phosphokinase (CPK)

## Data Availability

Not applicable.

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
