# Peer review of "Review of Immune-Related Adverse Events (irAEs) in Non-Small-Cell Lung Cancer (NSCLC)—Their Incidence, Management, Multiorgan irAEs, and Rechallenge"

_biomedicines, 2022, doi:10.3390/biomedicines10040790_

Round 1

Reviewer 1 Report

This is a very well-written manuscript with reader-friendly classification and manuscript skeleton. 

Minor comments:

Missing articles and silly grammatical errors: Authors need to extensively revise language for a proper sentence structure. 

e.g. 20 days following a single dose of nivolumab..........

6.10% for a severe grade...........

Management of all patients includes physical examinations......

Alternatively, treatment can be held until symptom........

microbiological workups ....

Species names are to have a capitalized genus and whole name in italics e.g.

Clostridioides (formerly Clostridium) difficile......

Authors mention role of mutations in irAEs once in introduction and there is no mention of recent vaccinations e.g. Flu shots on irAEs throughout the paper. Authors can dedicate a small para on predisposition factors which adds a translational aspect to the review

Authors have stated multiorgan irAEs are associated with good tumor response which suggests those irAEs are neoplastic syndrome associated. But there are numerous reports concluding that this phenomenon is more associated with the rate of irAEs.. This has enough studies for authors to discuss in detail. 

Major comments

A table with irAEs markers vs. markers  that  correlate  with  clinical  outcome  after  immune  checkpoint  inhibitors  for  non-small  cell  lung  cancer would greatly clarify the marker profile expected of a successful treatment. Currently, this information is too scattered and addressed non uniformly under different irAEs. The markers can be further Sub grouped into prognostic, predictive and (early) monitoring biomarkers within the table.

Authors need to summarize contraindications/markers for use of immune checkpoint inhibitors in NSLC. 

Genetic susceptibility to irAEs is completely ignored in the manuscript. Authors need to present the data in brief

The manuscript is very systematic but due to irAEs based classification a lot of biomarker information is either scattered or is repetitive. A tabular representation can be helpful in imbibing the whole picture and reduce repetitive information throughout the manuscript. 

Author Response

Reviewer 1 report:

This is a very well-written manuscript with reader-friendly classification and manuscript skeleton. 

We thank the reviewer for their time and valuable suggestions.

Minor comments:

Comment: Missing articles and silly grammatical errors: Authors need to extensively revise language for a proper sentence structure. 

Response: We have thoroughly revised the manuscript and corrected typographical and grammatical errors.

Comment: Authors mention role of mutations in irAEs once in introduction and there is no mention of recent vaccinations e.g., flu shots on irAEs throughout the paper. Authors can dedicate a small para on predisposition factors which adds a translational aspect to the review

Response: We have included a paragraph discussing mutation burden and risk of irAEs in the introduction part.  In addition, we also included a paragraph discussing predisposing factors and influenza vaccination.

Comment: Authors have stated multiorgan irAEs are associated with good tumor response which suggests those irAEs are neoplastic syndrome associated. But there are numerous reports concluding that this phenomenon is more associated with the rate of irAEs. This has enough studies for authors to discuss in detail. 

Response: We agree with reviewers that though multiorgan irAEs are associated with good tumor response the rate of irAEs is high. However due to our primary focus on specific irAEs we did not include detailed discussions on this topic.

Major comments

Comment: A table with irAEs markers vs. markers that correlate with clinical outcome after immune checkpoint inhibitors for non-small cell lung cancer would greatly clarify the marker profile expected of a successful treatment. Currently, this information is too scattered and addressed non uniformly under different irAEs. The markers can be further sub-grouped into prognostic, predictive and (early) monitoring biomarkers within the table.

Response: We have included a table summarizing management features of irAEs

Comment: Authors need to summarize contraindications/markers for use of immune checkpoint inhibitors in NSLC. 

Response: We have added a paragraph regarding contraindications and markers.

Comment: Genetic susceptibility to irAEs is completely ignored in the manuscript. Authors need to present the data in brief

Response: We have included information regarding genetic susceptibility in the introduction part.

Reviewer 2 Report

I enjoyed reviewing this manuscript. The paper is interesting and well written. The topic is very hot.

This reviewer only recommends a few tips.

1- Lines 390-392. Authors state “For patients with new-onset hyperglycemia less than 200 mg/dL and history of T2DM with low suspicion for DKA, ICIs can be continued with serial blood glucose monitoring at each dose.” It is evident that, assessing the cost-benefit ratio, acceptable side effects such as those mentioned above should not impose changes in ICI therapy. However, it is equally true that elevated blood glucose values, even in subjects without known diabetes, must be carefully monitored and adequately treated with adequate therapy, in case these patients develop acute coronary syndrome or due to ICI therapy itself (lines 556) or as a comorbidity. Indeed, tight glycemic control during acute coronary syndrome improves CV outcome and mortality in diabetic and non-diabetic patients (Journal of Clinical Endocrinology and Metabolism Volume 97, Issue 3, March 2012, Pages 933-942. Doi: 10.1210 / jc.2011-2037 - Journal of Diabetes Research, 2018, art.no 3106056. doi: 10.1155 / 2018/3106056). This hot issue should be commented on with the references above.

2- There is only one introductory figure in the text. I believe it would be very helpful to the reader for the authors to add 1 or 2 tables summarizing the immune-related adverse events described in the review.

Author Response

Reviewer 2 Report:

Comment: The manuscript is very systematic but due to irAEs based classification a lot of biomarker information is either scattered or is repetitive. A tabular representation can be helpful in imbibing the whole picture and reduce repetitive information throughout the manuscript. 

Response: We thank the reviewer for their time and valuable suggestions. A comprehensive table has been added as per reviewer suggestions.  

I enjoyed reviewing this manuscript. The paper is interesting and well written. The topic is very hot.

This reviewer only recommends a few tips.

Comment: Lines 390-392. Authors state “For patients with new-onset hyperglycemia less than 200 mg/dL and history of T2DM with low suspicion for DKA, ICIs can be continued with serial blood glucose monitoring at each dose.” It is evident that, assessing the cost-benefit ratio, acceptable side effects such as those mentioned above should not impose changes in ICI therapy. However, it is equally true that elevated blood glucose values, even in subjects without known diabetes, must be carefully monitored and adequately treated with adequate therapy, in case these patients develop acute coronary syndrome or due to ICI therapy itself (lines 556) or as a comorbidity. Indeed, tight glycemic control during acute coronary syndrome improves CV outcome and mortality in diabetic and non-diabetic patients (Journal of Clinical Endocrinology and Metabolism Volume 97, Issue 3, March 2012, Pages 933-942. Doi: 10.1210 / jc.2011-2037 - Journal of Diabetes Research, 2018, art.no 3106056. doi: 10.1155 / 2018/3106056). This hot issue should be commented on with the references above.

Response: We agree with the reviewer that tight glycemic control is important even in patients without know history of diabetes. We have made changes based on reviewer suggestions. 

Comment: There is only one introductory figure in the text. I believe it would be very helpful to the reader for the authors to add 1 or 2 tables summarizing the immune-related adverse events described in the review.

Response: A comprehensive table on management of irAEs has been added as per reviewer suggestions. 

Round 2

Reviewer 1 Report

All queries have been satisfactorily addressed